# Dynamics of pattern formation and emergence of swarming in *Caenorhabditis elegans*

Esin Demir[1†], Y Ilker Yaman[2†], Mustafa Basaran[1], Askin Kocabas[1,2,3,4]*

[1]Bio-Medical Sciences and Engineering Program, Koç University, Sarıyer, Istanbul, Turkey; [2]Department of Physics, Koç University, Sarıyer, Istanbul, Turkey; [3]Koç University Surface Science and Technology Center, Koç University, Sarıyer, Istanbul, Turkey; [4]Koç University Research Center for Translational Medicine, Koç University, Sarıyer, Istanbul, Turkey

**Abstract** Many animals collectively form complex patterns to tackle environmental difficulties. Several biological and physical factors, such as animal motility, population densities, and chemical cues, play significant roles in this process. However, very little is known about how sensory information interplays with these factors and controls the dynamics of pattern formation. Here, we study the direct relation between oxygen sensing, pattern formation, and emergence of swarming in active *Caenorhabditis elegans* aggregates. We find that when thousands of animals gather on food, bacteria-mediated decrease in oxygen level slows down the animals and triggers motility-induced phase separation. Three coupled factors—bacterial accumulation, aerotaxis, and population density—act together and control the entire dynamics. Furthermore, we find that biofilm-forming bacterial lawns including *Bacillus subtilis* and *Pseudomonas aeruginosa* strongly alter the collective dynamics due to the limited diffusibility of bacteria. Additionally, our theoretical model captures behavioral differences resulting from genetic variations and oxygen sensitivity.

*For correspondence:
akocabas@ku.edu.tr

†These authors contributed equally to this work

Competing interests: The authors declare that no competing interests exist.

## Introduction

Some animals have remarkable abilities to form complex patterns to cope with environmental challenges (*Sumpter, 2010*; *Martínez et al., 2016*; *Miller and Gerlai, 2012*; *Szopek et al., 2013*; *Ioannou et al., 2012*; *Stabentheiner et al., 2003*; *Peleg et al., 2018*; *Tennenbaum et al., 2016*; *Gregor et al., 2010*). Formation of these biological patterns is particularly initiated by aggregation. At high population density, animals tend to aggregate as animal-to-animal interactions cause instability in uniform distribution (*Gregor et al., 2010*; *Buhl et al., 2006*; *Liu et al., 2016*; *Vicsek et al., 1995*). Different factors such as motility of animals and chemical cues can trigger these instabilities by altering the behavior of animals. In particular, sensory information plays a primary role in controlling behaviors. Elucidating how environmental factors interplay with the sensory information is essential to improve our understanding of pattern formation in biological systems. However, this is a challenging problem in complex organisms.

Nematode *Caenorhabditis elegans* allows the use of sophisticated experimental tools and the results provide very precise information regarding the relation between genes, neural circuits, and behaviors (*Bargmann and Marder, 2013*; *Gray et al., 2005*; *Chalasani et al., 2007*; *Calarco and Samuel, 2019*; *Kocabas et al., 2012*; *Lee et al., 2019*). This study is motivated by the possibility of developing a platform that combines both neuronal information and the collective dynamics in this model system.

*C. elegans* shows a variety of collective responses including, social feeding (*de Bono and Bargmann, 1998*; *de Bono et al., 2002*; *Macosko et al., 2009*; *Coates and de Bono, 2002*), starvation-

induced clustering (*Artyukhin et al., 2015*), hydrodynamic networking (*Sugi et al., 2019*) and collective swimming (*Yuan et al., 2014*; *Gray and Lissmann, 1964*). Recent studies have shed light on some physical (*Sugi et al., 2019*; *Ding et al., 2019*; *Oda et al., 2017*) and biological (*Rogers et al., 2006*; *Busch et al., 2012*; *Zhao et al., 2018*) aspects of these behaviors and their variations. Oxygen sensitivity is particularly the main source of the variation. Laboratory strain (N2) is known to be solitary nematode with a broad range of oxygen preference while natural isolates exhibit strong aggregation and sharp aerotaxis (*Rogers et al., 2006*; *Persson et al., 2009*). Further, oxygen sensing, feeding and aggregation behavior are all intricately related in *C. elegans*. Bacteria utilize oxygen for growth and *C. elegans* seeks low oxygen levels for locating the bacteria as a food source. However, in a dense population, all these factors become extremely coupled (*Tuval et al., 2005*; *Gray et al., 2004*; *Nichols et al., 2017*). In this study, we aimed to develop a general model dealing with the dynamics of all these factors.

Here, we report that, *C. elegans* including the laboratory strain (N2), forms complex patterns during feeding. When thousands of worms are forced to feed together, aggregation-induced bacterial accumulation and oxygen depletion create unstable conditions and further trigger phase separations. The principle of phase separation is mainly based on a sudden change in the animal's motility (*Liu et al., 2016*; *Cates and Tailleur, 2015*). We also found that the dynamics of the entire process is controlled by the sensitivity of the oxygen-sensing neurons, which gives rise to strong variations in animals' collective response. Further, we also observe the convergence of self-organized patterns to the motile swarming body as a result of complex interactions between oxygen diffusion, bacterial consumption, and motility of animals. Finally, we studied the effects of bacterial diffusibility and biofilm formation of the lawn on the collective dynamics of animals.

## Results

To investigate the collective response of *C. elegans,* we began by imaging animals during food search in a crowded environment. We collected thousands of animals (N2) in a droplet and let it dry on an agar plate. The droplet was put a few centimeters away from food. After drying, worms spread and searched for food by tracing the attractive chemicals. This experimental procedure allowed us to increase the worm density around a bacterial lawn to observe their collective movement. When animals found the bacteria, they usually slowed down and penetrated the lawn. However, we observed that, at high densities the majority of the worms did not penetrate the lawn. After reaching the lawn, they suddenly stopped and started accumulating (*Figure 1a*). As the worms accumulated, they formed a huge aggregate that covered the entrance. Eventually, this large aggregate gained motility and swarmed across the lawn (*Figure 1a,b,c*, *Video 1*). This observation was quite different from the previously reported response of the reference strain N2 which generally behaves solitarily (*de Bono and Bargmann, 1998*; *Gray et al., 2004*). Further, we tested different mutant strains deficient in several sensing mechanisms and essential neuromodulators; we observed similar aggregation and swarming responses in a wide range of animals (*Figure 1—figure supplement 1*). We also used various types of food sources including *B. subtilis* biofilm, filamentous bacteria, or extremely thick bacterial lawns. Surprisingly, animals showed a very rich collective response ranging from complex pattern formations to large scale swarming (*Figure 1d,e,f*, *Figure 1—figure supplement 2*). Altogether, these results suggest that the conditions observed in a dense population can broadly trigger the collective response in *C. elegans*.

This collective response only occurs on a bacterial lawn. Previous studies reported the same conditions for aggregating strains (*de Bono and Bargmann, 1998*; *Gray et al., 2004*). However, how the presence of bacteria contributes to this process is unclear. To clarify this point, we used GFP expressing bacteria (OP50, *E. coli*) to follow the entire dynamics. Time-lapse fluorescence microscopy revealed that the bacteria are concentrated within the aggregates. This bacterial accumulation further promotes the collective behavior of animals (*Figure 1f*, *Figure 1—figure supplement 3*, *Figure 1—video 1*; *Video 2*). Following the accumulation of bacteria, the motility of animals is strongly suppressed. The naïve hypothesis explaining this observation is that the process is mainly based on the capillary meniscus around the animals (*Rabets et al., 2014*; *Gart et al., 2011*). When animals form aggregates, the structure becomes porous. Thus, due to the capillary effect, the porous aggregate can hold more bacterial suspension. Eventually, concentrated bacteria could make the formation of aggregate more favorable by conditioning the oxygen levels. We can conclude that

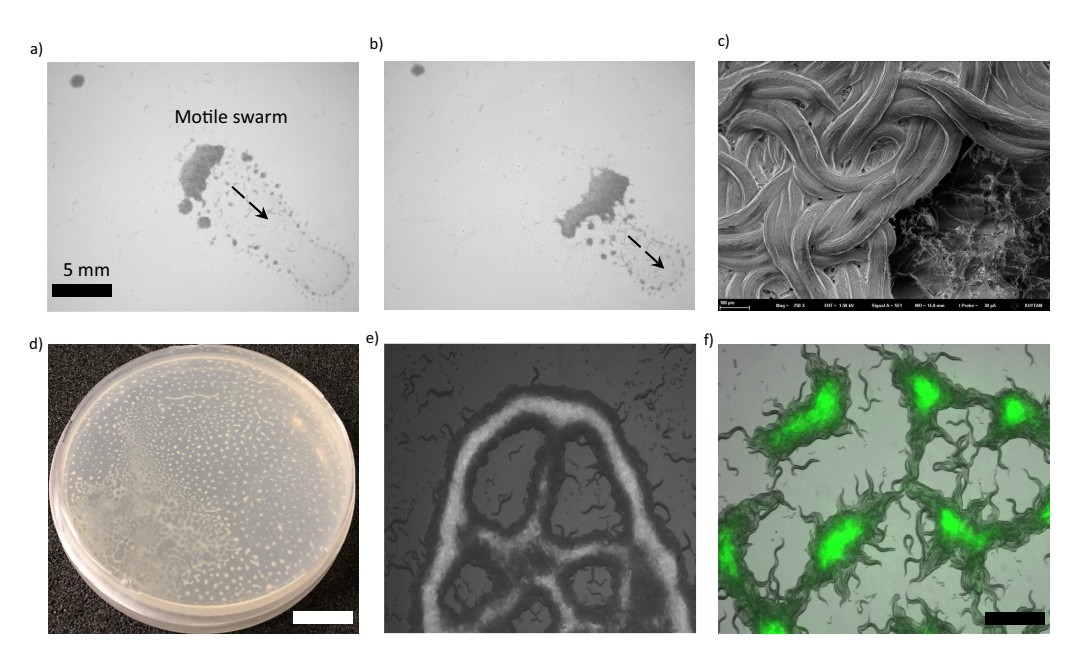

**Figure 1.** Collective response and pattern formations in *C. elegans*. (**a, b**) Thousands of animals collectively swarm on a bacterial lawn. (**a**) While searching for food, worms aggregate around the edge of the bacterial lawn. Small aggregates merge and form a massive swarm. (**b**) Swarming body eventually gains motility and moves across the lawn while consuming the bacteria. (**c**) A cryo-SEM image of the swarming body. The heads and the tails of the animals are buried, and the bodies extend on the surface of the swarm. (**d, e**) Sample images of Turing-like patterns formed by animals on an agar surface under various conditions, such as starvation-induced and on a biofilm. The magnified version of the image is given in the supplementary information. Scale bar, 1 cm (**f**) Fluorescent image of worms aggregating near the GFP-labeled bacteria. Due to the capillary effect and diffusion of bacteria, worms concentrate the bacteria within the aggregates. Scale bar, 1 mm.

The online version of this article includes the following video and figure supplement(s) for figure 1:

**Figure supplement 1.** Average swarm area formed by different mutant strains.

**Figure supplement 2.** Sample patterns formed under different bacterial conditions.

**Figure supplement 3.** Worm density and bacterial concentration.

**Figure 1—video 1.** Emergence of swarming response of *npr-1* mutant around a GFP labeled bacterial lawn.

https://elifesciences.org/articles/52781#fig1video1

**Figure 1—video 2.** Response of the swarm to the placement of cover glass.

https://elifesciences.org/articles/52781#fig1video2

**Figure 1—video 3.** Measurement of $O_2$ diffusion on an agar surface.

https://elifesciences.org/articles/52781#fig1video3

**Figure 1—video 4.** Accumulation of worms around $Na_2SO_3$ containing agar chunk.

https://elifesciences.org/articles/52781#fig1video4

**Figure 1—video 5.** Response of npr-1 animals on very thick bacterial lawn.

https://elifesciences.org/articles/52781#fig1video5

concentrated bacteria is the primary factor triggering the formation and the maintenance of the aggregation.

We then focused on the effects of oxygen on the entire process. Previous studies have already shown that the bacterial lawn can decrease oxygen concentration [$O_2$] (*Rogers et al., 2006*; *Gray et al., 2004*). To directly observe this effect, we used fiber optic sensors to measure [$O_2$] in both bacterial suspensions and in the swarming body. We observed that the oxygen level drops below 0.1% in the bacterial suspension (*Figure 2—figure supplement 1*). Moreover, we found similar depletion in [$O_2$] in the swarming body (*Figure 2a*, *Video 3*). These results imply that concentrated bacteria deplete [$O_2$] in the swarming body. It should be noted that the fiber optic sensor can only be used to record the oxygen levels in the swarm liquid. Although [$O_2$] is very low, the animals' bodies can extend on the surface of the swarm and they can get sufficient $O_2$ from the environment. Besides the ambient environment, wet agar surface can also provide oxygen by lateral surface

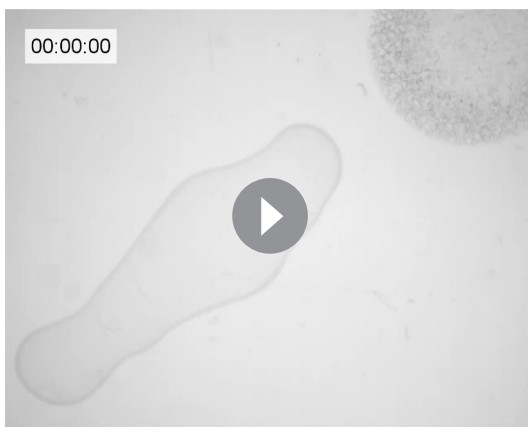

**Video 1.** Typical swarming response of a laboratory strain (N2) *C.elegans*. The video is played 600 X speed and the real elapsed time is indicated in the video (h: min: sec). This video is associated with *Figure 1a,b*.
https://elifesciences.org/articles/52781#video1

diffusion. We independently validated these contributions by covering the aggregate with a glass slide and by using an agar chunk containing oxygen depleting chemical $Na_2SO_3$, (*Figure 2—figure supplement 2*, *Figure 1—video 2 and Figure 1—video 3* ). Moreover, we observed that $Na_2SO_3$ based oxygen depletion on the agar surface could initiate aggregation behavior in the bacteria-free region (*Figure 1—video 4*). We also tested animals responses on a very thick bacterial lawn which can significantly decrease the oxygen level. We found that *npr-1* worms did not form clusters on the thick bacterial lawn and they avoided extremely thick regions of the lawn (*Figure 1—video 5*). As a final control experiment we tested the aggregation responses of worms on a dead OP50 lawn. Interestingly, dead bacterial suspensions have different oxygen kinematics and result in small clustering (more discussions about dead bacteria are available in the review file). From these results, we can conclude that animals effectively experience the average $O_2$ levels defined by the ambient environment and the swarm liquid. Altogether, ambient oxygen levels and oxygen depletion are essential factors for aggregation.

To quantify the effects of $[O_2]$ on the motility of animals, we measured the response of the animals under changing oxygen concentrations. *Figure 2b* compares the velocity profiles V(O) of both, aggregating strain *npr-1* (DA609) and solitary strain N2 and the velocity of the animals moving in the bacteria-free region (off-food). Both strains responded to very low $[O_2]$ by increasing their velocities, which mainly triggers dispersive behavior (*Video 4*). On the contrary, *npr-1* suppressed its motility around intermediate oxygen levels (7–10%) and showed a sharp response when oxygen levels exceeded 15%. In contrast, N2 suppressed its motility in more broader range of oxygen levels, but their velocity slowly increased as $[O_2]$ reached 21% or more. These differences appear to be originating from the sensitivity of the oxygen-sensing neurons URX, AQR and PQR that shape the overall oxygen preferences of the animals (*Oda et al., 2017*; *Busch et al., 2012*; *Zimmer et al., 2009*). Although both *npr-1* and N2 perform aerotaxis at 7–10% oxygen, N2 shows a weaker aerotactic response, thus, they have a broader range of oxygen preference. All these critical features of the strains can be extracted from oxygen-dependent response curve V(O) which is directly related to neuronal sensitivity.

Motility suppression behavior has been observed in a variety of organisms ranging from bacteria to mussels (*Cates and Tailleur, 2015*; *Cates et al., 2010*; *Liu et al., 2013*; *Liu et al., 2014*; *Dervaux et al., 2017*; *Liu et al., 2011*; *Garfinkel et al., 2004*). Generally, animals tend to slow down when they come together. The entire process is represented by the density dependence of the animal movement which can lead to the formation of patterns. However, in *C. elegans* we observed indirect density-dependent

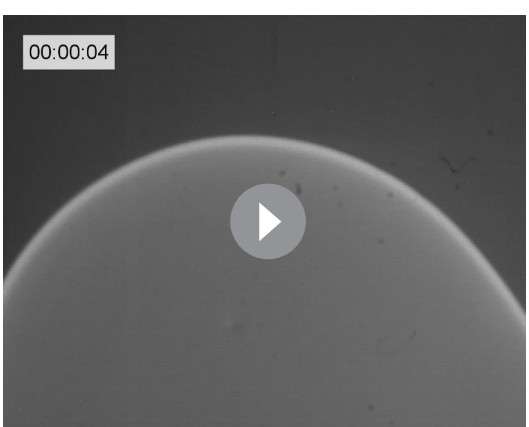

**Video 2.** Emergence of swarming response of laboratory strain (N2) around a GFP-labeled bacterial lawn. Worms accumulate around the lawn and form large aggregates. As large aggregates merge, swarm body appears and gains motility. Bacteria accumulate within the aggregates. The video is played 250 X speed and the real elapsed time is indicated in the video (hr: min: s).
https://elifesciences.org/articles/52781#video2

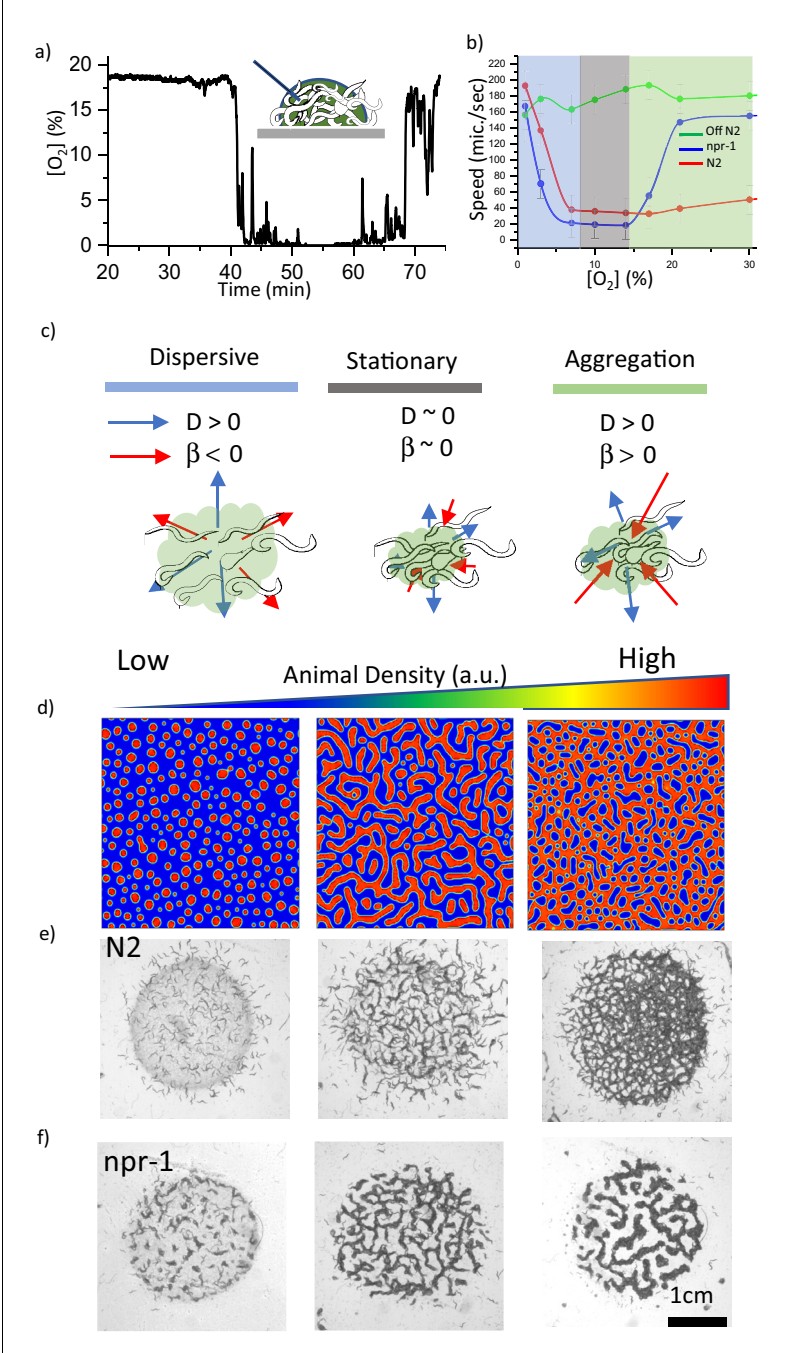

**Figure 2.** Oxygen sensitivity and the dynamics of pattern formation. (**a**) Accumulation of bacterial suspension depletes oxygen in worm aggregates. Oxygen levels measured by the fiber optic sensor while the swarm passes through the sensor. Initially fiber optic sensor records ambient [$O_2$] on the surface and the sudden drop is originating from the complete coverage of the fiber by the swarm body. (**b**) Oxygen-dependent motility of animals. The average speed of the worms as a function of ambient oxygen. The slope of the curve defines the aerotactic response of the animals. The comparison of velocity profiles of N2 (red) (on food), npr-1(blue) (on food), and off-food-N2 (green) response show the difference between the strains. Colored regions indicate the dispersive, stationary and aggregation phases. Error bar indicates the s.e.m. = ± 18 µm/s. For each experiment, 25 individual animals were imaged. Measurement error is defined by the error in image processing steps. (**c**) Schematic representation of each phase. The competition between animal dispersion (**D**) and aerotactic response (β) defines the collective behavior of worms. (**d**) Simulation results of the model at time 10 min after randomization of the patterns with various initial animal densities. The pattern formation strongly depends on worm density. At

*Figure 2 continued on next page*

*Figure 2 continued*

low density, worms create clusters. As the worm density increases, clusters converge to stripes and hole patterns. (e, f) Experimental results of density-dependent pattern formation. (e) At low worm densities (500 worms/cm²), N2 worms do not form patterns. As the density increases, we observed stripes (2000 worms/cm²) and holes (6000 worms/cm²). (f) The strong aerotactic response of npr-1 strain enables worms to form patterns even at low densities. An increase in the worm density causes a change in the instability conditions and tunes the patterns. Scale bar, 1 mm.

The online version of this article includes the following figure supplement(s) for figure 2:

**Figure supplement 1.** Oxygen measurement in a bacterial suspension[O₂] in a bacterial suspension as a function of time (in blue).

**Figure supplement 2.** Contributions of surface penetration and lateral diffusion of oxygen.

**Figure supplement 3.** Effects of thick bacterial lawn.

**Figure supplement 4.** Imaging system.

---

suppression. Without bacteria, animals move fast (*Figure 2b*). In striking contrast, the presence of bacteria results in oxygen depletion and motility suppression. These findings suggest that in the dynamics of oxygen, bacterial concentration and animal motility are the essential physical factors controlling the collective behaviors of *C. elegans*.

To gain a more quantitative representation of pattern formation, we developed a mathematical model. Many different models have been used to describe the dynamics of pattern formation. Particularly, dryland vegetation models share similarities in terms of dynamic variables and feedbacks (*Gilad et al., 2007*; *Gilad et al., 2004*). However, in our system, we have to implement the activity of the worms based on oxygen concentrations. To do so, we followed the notation and the framework developed for active chemotactic particles (*Liebchen and Löwen, 2018*). We set two separate differential equations to represent worm density (W) and oxygen kinematics (O) in two-dimensional space. V(O) is the oxygen-dependent motility response of individual animals. This factor served as an experimentally measurable sensory curve of the animals. This curve defines competition between diffusivity and aerotactic motility. Both effects became minimal around the optimum oxygen level (7–10%) at which the animals were almost in a stationary phase (*Figure 2b*). In low oxygen region, dispersion and reversal of aerotaxis promoted motility. On the other hand, at high oxygen levels, aerotaxis dominated the dynamics and promoted aggregation (*Figure 2c*). These are the Keller-Segel like equations (*Keller and Segel, 1970*; *Wang, 2010*) with both, nonlinear diffusion and chemotactic sensitivity (Materials and methods);

$$\frac{\partial W}{\partial t} = \nabla[D_W \nabla W] + \nabla[\beta W \nabla O] \tag{1}$$

$$\frac{\partial O}{\partial t} = D_O \nabla^2 O + f(O_{am} - O) - k_c W \tag{2}$$

Here, $D_W = \frac{V^2}{2\tau}$ represents motility-dependent dispersion of the worms, $D_O$ is the diffusion coefficient of oxygen on the surface, $\beta = \frac{V}{2\tau}\frac{\partial V}{\partial O}$ is the aerotactic coupling coefficient indicating the strength of the animal response to oxygen gradient. $f$ is the penetration rate of the oxygen from the air to the water surface, $O_{am}$ is the ambient oxygen level, and $k_c$ is the rate of oxygen consumption by worms and bacteria. $k_c W$ term in the model covers the hydrodynamics related to the bacterial concentration. Here, we assume that bacterial density is linearly proportional to worm density (*Figure 1—figure supplement 3b*). Furthermore, the time scale of the bacterial consumption is much longer than the

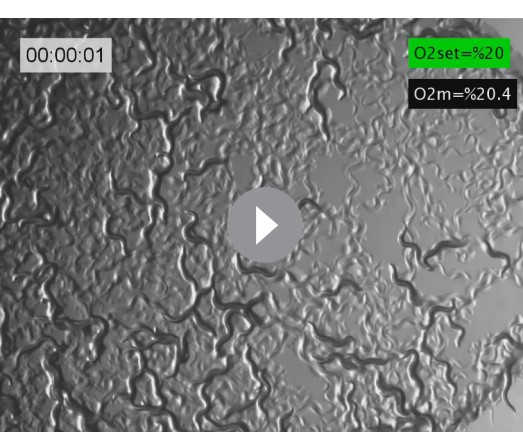

**Video 3.** Measurement of [O₂] in a swarming body. The fiber optic sensor is placed on a bacterial lawn and ambient [O₂] is set to 21%. Ambient set [O₂] and measured [O₂] are indicated in the video as O2 set (green) and O2m (black), respectively. This video is associated with *Figure 2a*.

https://elifesciences.org/articles/52781#video3

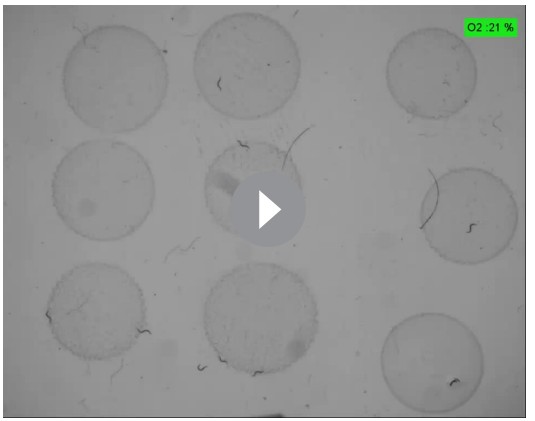

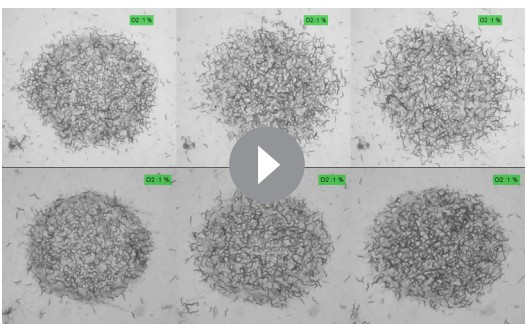

**Video 4.** Oxygen-dependent motility response of individual animals (npr-1). In order to minimize the animal to animal interaction isolated bacterial slots were used. This video is associated with *Figure 2b*. https://elifesciences.org/articles/52781#video4

**Video 5.** Typical structural patterns formed by the animal at high population density under different oxygen levels. Top images show N2 and bottom images corresponds to npr-1 strain. The circular region is the bacterial lawn. This video is associated with *Figure 2d–f* and *Figure 3*. https://elifesciences.org/articles/52781#video5

aggregation time, thus we ignore the bacterial consumption term in the equations.

First, we focused on the instability criteria. Following the linear stability analysis of the solution in the uniform phase, we predicted that instability occurs when $W_{eq}\beta k_c > fD_W$ . The detailed derivations are given in Materials and methods. This criterion suggests that population density and aerotactic response favor instability; however, dispersion of animals opposes it. To verify these conditions, we performed simulations by tuning initial worm density. Depending on the density, we observed the formation of uniform distribution, dots, stripes, and holes (*Figure 2d*). These are the basic patterns that are frequently seen in many biological systems (*Liu et al., 2016*; *Chen et al., 2012*).

Our simulations also predict that under low aerotactic response, the increase in population density could complement the instability criteria and give rise to pattern formation. This is because the multiplication of density and aerotactic response simply shifts the dynamics to a new instability zone. This prediction could explain the response of solitary N2 in a dense population. N2 shows weak aerotactic response which results in broad oxygen preference and patterns only appear in a dense environment. On the other hand, the strain *npr-1* with sharp aerotactic response can form similar patterns even at low population density. We tested this hypothesis experimentally (*Figure 2e and f*, *Videos 5* and *6*) and found that N2 forms stripes and hole-like patterns only at high population density, whereas *npr-1* could form small aggregation patterns even at a low density.

Ambient oxygen is the second experimentally controllable factor driving the dynamics of the system. The onset of the pattern formation can be controlled by tuning the ambient oxygen levels at a fixed population density. Further, we tested the contribution of ambient oxygen levels to pattern formation. To be able to control the [O$_2$], we performed the experiments in a chamber where the flow of O$_2$/ N$_2$ mixture is precisely controlled (*Figure 2—figure supplement 3*). First, we measured the [O$_2$] in a large aggregate accumulated around the bacterial lawn. We started the experiment at 21% O$_2$ and sequentially decreased the [O$_2$] to 1%. We observed that the aggregates respond to decreasing O$_2$ level by increasing their surface area (*Figure 3a*, *Figure 3—figure supplement 1*, *Video 7*). However, [O$_2$] in the aggregate remained low until it reached 7% (*Figure 3b*). Below this level, the pattern became very porous and oxygen would directly leak into the fiber sensor which led to small fluctuations in readings. It should be noted that the [O$_2$] at 1% completely randomized the patterns by increasing motility and reversing aerotaxis. Subsequently, we increased the [O$_2$] which resulted in similar stripe and hole patterns across the region (*Figure 3c*, *Figure 3—figure supplement 2*). We noticed that patterns formed by N2 strain did not change significantly during oxygen scans (*Figure 3—figure supplement 3*). This was expected, because N2 exhibits low and almost flat aerotactic response within the same range of oxygen levels.

We further explored the combinatorial effects of oxygen and worm density on pattern formation. As predicted by the instability criterion, in two-dimensional parameter space, the onset of dot, stripe

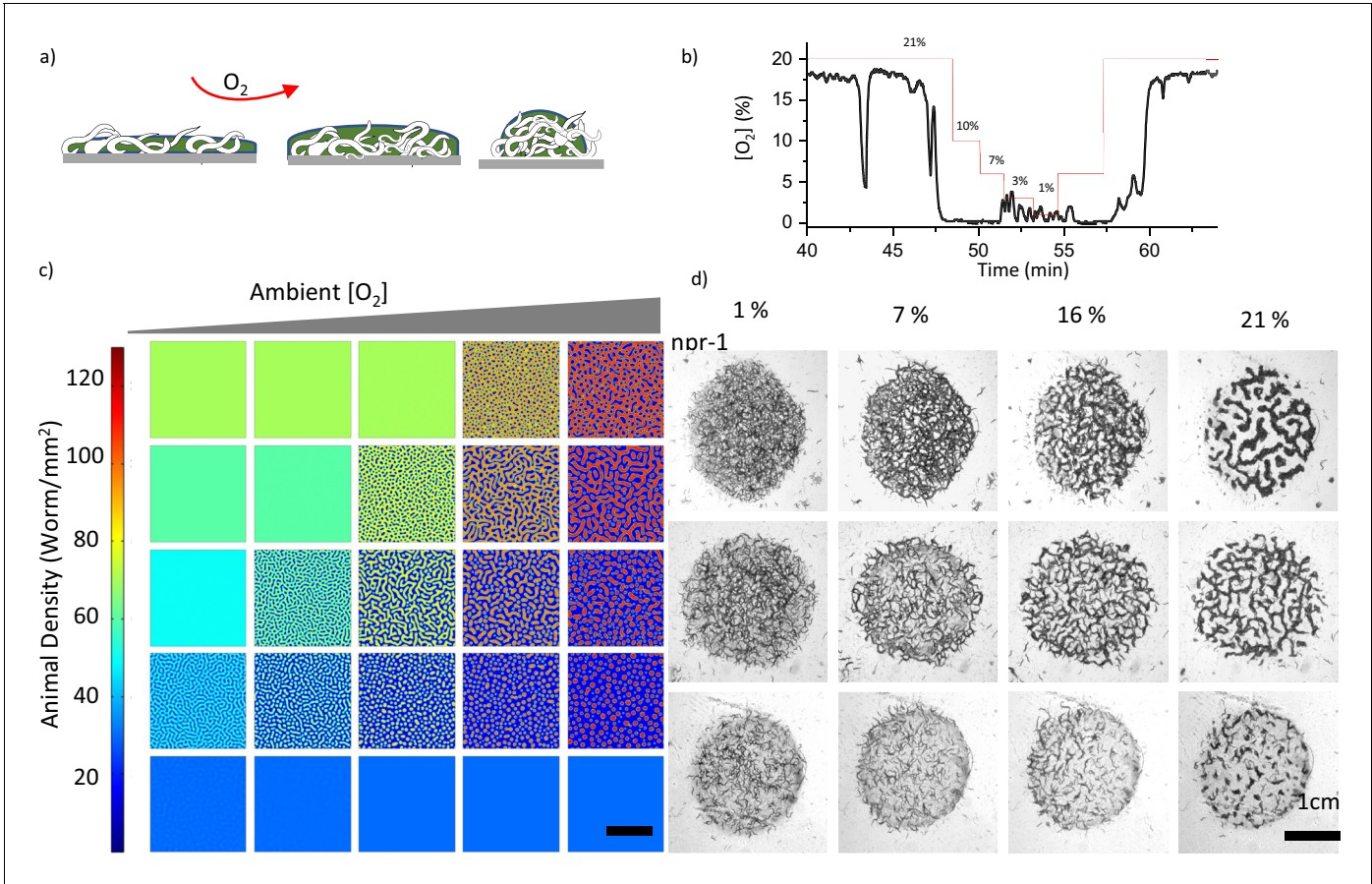

**Figure 3.** Combinatorial effect of ambient oxygen and worm density. (**a**) Schematics show the response of worm aggregates to changing ambient oxygen levels. As oxygen level increases, the size of the clusters shrink and form circular aggregates which balance the oxygen penetration and diffusion. (**b**) Measured oxygen concentration in worm aggregates. The dashed red line shows the ambient oxygen level. Error bar indicates the s.e. m. = ± 0.1% (**c, d**) Simulation and experimental results of pattern formation under various worm densities and ambient oxygen levels. Patterns are formed when the instability criterion is satisfied. The instability zone is bounded by the uniform stable population. Scale bar, 1 cm. Image snapshots representing simulation results are taken at time 10 min after randomization of the patterns. Worm densities vary between 5–60 worms / mm² (*Martínez et al., 2016*) in the experiments.

The online version of this article includes the following figure supplement(s) for figure 3:

**Figure supplement 1.** Response of the swarm to changing oxygen.
**Figure supplement 2.** Sample patters formed by npr-1, under different oxygen levels.
**Figure supplement 3.** Sample patterns formed by N2, under different oxygen levels.
**Figure supplement 4.** Simulation results of worm density and oxygen profiles through the crossection of the clusters.

and hole patterns is observed around the zone bounded by high and low uniform animal densities (*Figure 3d*). An increase in worm density or a decrease in ambient oxygen levels transforms the dot-shaped structure to stripe and hole patterns.

The other interesting feature of the pattern formation is the coarsening event. We sought to know how the shape of the patterns evolves in time at a fixed oxygen level. Our time-lapse imaging and simulation results validate the coarsening of characteristic domain size (*Bray, 1994*; *Figure 4a–c*, *Figure 4—figure supplement 3*, *Videos 8,9*). In later stages, patterns merge and form a large cluster. This type of a large cluster shares visual similarities with our initial swarming experiments (*Figure 1a and b*). Interestingly, unlike simulation the width of the of the cluster reaches saturation before they dissociate. Arrested phase seperation was also observed in bacterial system due to birth and death dynamics (*Cates et al., 2010*). We noticed that, in our case, the consumption of bacteria is significant within the cluster and may limit the growth of the clusters. The similar effect of bacterial depletion was also proposed to explain the motion of the aggregating strain *npr-1* (*Ding et al., 2019*). To investigate the details, we measured the bacterial concentration. GFP signal revealed

different bacterial concentrations across the swarm; the front edge of the swarm has more bacteria than the back (*Figure 4d–f*). Worms in the swarm consumed bacteria and the food continually diffused from the front edge toward the back. As the swarm grows, the gradient profile gradually extends into the swarming body with the average decay length of around $\lambda \sim 2–4$ mm (*Figure 4g–h*). $\lambda$ the characteristic length scale of the gradient profile can be simply defined by the bacterial diffusion coefficient ($D_b$) and consumption rate ($f_b$), $\lambda^2 = D_b/f_b$. This decay length defines the width of the swarm body. The small clusters show symmetric GFP distribution and do not form a gradient profile (*Figure 4—figure supplement 1d*).

Next, we tested whether this concentration profile could change the activity of the animals. The activity of the animals increased towards the back which suggests that the animals crawling at the front edge encountered more bacteria than those at the back. To quantify the activity profile, we measured the mean velocity of animals using Particle Image Velocimetry (PIV) analysis. Indeed, the velocity increased toward the back (*Figure 4—figure supplement 1a–c*). This response is consistent with our V(O) curve where the animals perform off-food response. Without the availability of bacteria, animals start moving fast. On the whole, due to the balance between bacterial consumption and diffusion, swarm body gains motility and moves across the bacterial lawn (*Figure 4—figure supplement 2*).

So far we have studied the interplay between oxygen levels and animal density. Our observations suggest that diffusion of bacteria on the lawn and swarm fluid is an essential factor and physical properties of the bacterial lawn could also play a significant role during pattern formation. To gain more understanding, we tested the responses of the animals on different biofilm-forming bacterial lawns. *B. subtilis* can form a biofilm which consists of an elongated chain of cells. Due to their long chaining structure, diffusibility of the bacteria could be relatively limited. We used GFP-labeled *B. subtilis* and overexpressed sinI protein (*Yaman et al., 2019*) to drive biofilm formation on regular NGM plates. *Figure 5* shows the time evolution of animal accumulation on this chaining biofilm. Interestingly, as animals accumulate dark regions appear across the lawn. These regions indicate the bacteria depleted domains that eventually grow and form open holes (*Video 10*). The appearance of these holes triggered by local bacterial depletion which can be considered as noise in the concentration. These structures *Tjhung et al., 2018* strongly resemble the noise-induced bubbly phase seperation (*Tjhung et al., 2018*) in active particles. In order to get more insights we tested the dynamics of holes in a swarming body moving on isolated *E. coli* lawn (OP50). Interestingly, on *E. coli* based swarm, holes gradually disappeared (*Figure 5—figure supplement 1*). This difference suggests that the dynamics of holes depend on the physical properties of the bacterial lawn. This difference could be related to the reversal of Ostward process observed in active fluids (*Tjhung et al., 2018*). We leave the details of this process for future work. The naïve explanation behind this process is that on biofilms the diffusibility of the

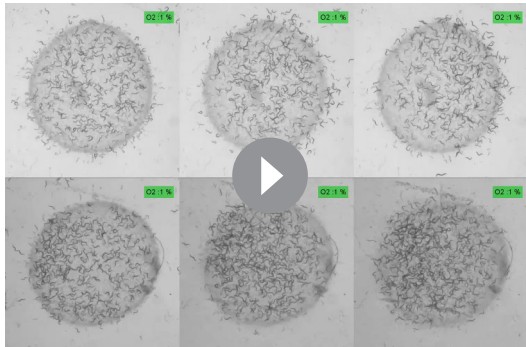

**Video 6.** Typical structural patterns formed by the animal at low population density under different oxygen levels. Top images show N2 and bottom images corresponds to npr-1 strain. The circular region is the bacterial lawn. This video is associated with *Figure 2d–f* and *Figure 3*.
https://elifesciences.org/articles/52781#video6

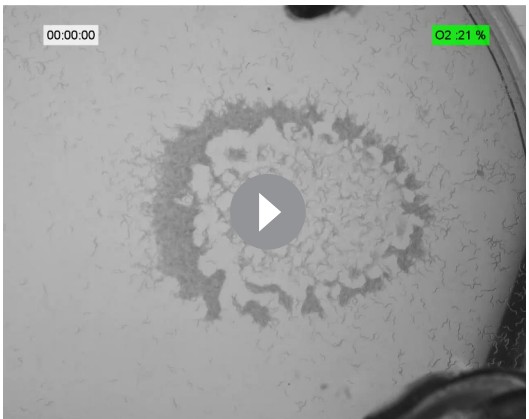

**Video 7.** Typical response of the swarm formed by *N2* to changing ambient oxygen.
https://elifesciences.org/articles/52781#video7

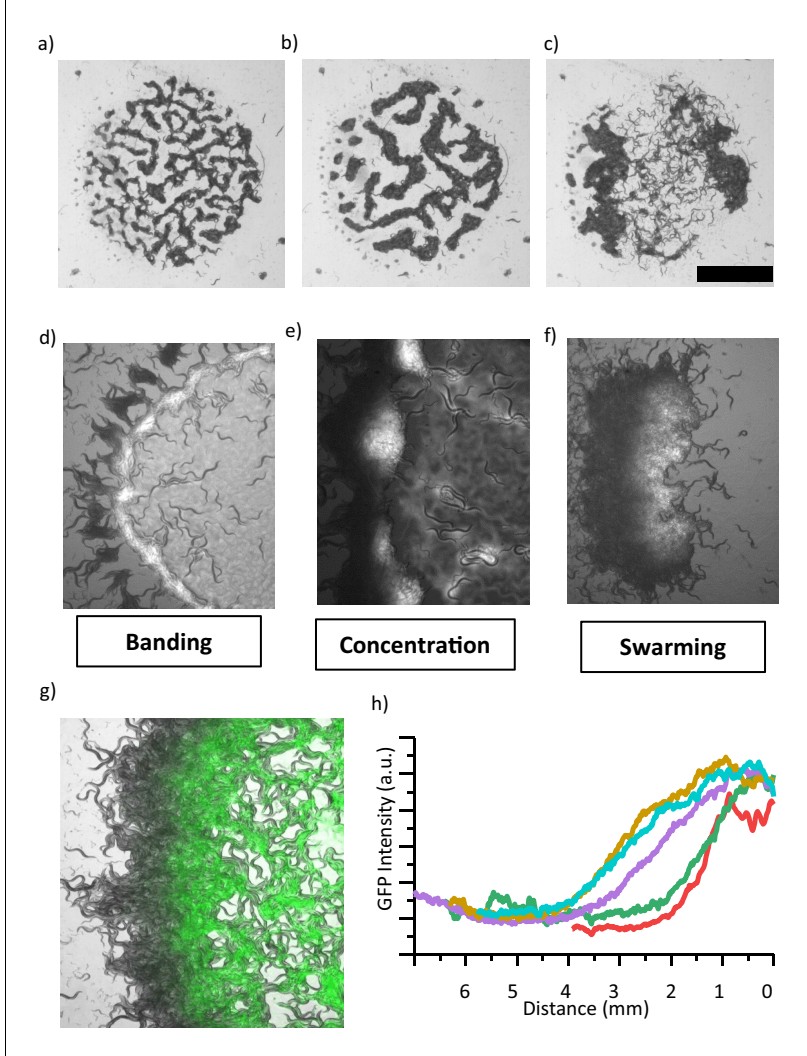

**Figure 4.** Coarsening event and the emergence of swarming behavior. (**a, b, c**) After the formation of patterns, animals locally consume bacteria which eventually increases the motility of animals. (**b**) Further, patterns merge and coarsen. (**c**) Consumption of food eventually results in the formation of a large aggregate which resembles the swarming body. Scale bar, 1 cm. (**d, e, f**) Different stages of a swarming response observed during food search behavior in a dense population. (**d**) Worms accumulate around the lawn. Unlike isolated worms, diffused bacteria cause the worms to slow down and then the worms form a band around the edge of lawn which eventually converges to aggregates. (**e**) Small aggregates concentrate bacteria and further accumulate the incoming worms. Consumption and diffusion of bacteria from the lawn leads to a bacterial gradient within the aggregate. (**f**) Following the formation of a gradient, large aggregate gains motility. The swarm moves through the lawn until finishing the entire lawn and then dissociates. (**g**) Superposition image of swarming animals and GFP-labeled bacteria indicating the gradient formation across the swarm. (**h**) A gradient profile gradually extends into the swarm at different time points.

The online version of this article includes the following figure supplement(s) for figure 4:

**Figure supplement 1.** Gradient profile of bacterial concentration.

**Figure supplement 2.** Center of mass velocities of the swarm (blue) and small aggregates (colored) as a function of time.

**Figure supplement 3.** Coarsening dynamics of both simulation and experimentally measured worm clusters in (**a**) logarithmic and (**b**) linear plots.

bacteria decreases and could not compensate the local oxygen penetration which further promotes animal motility around these depleted domains.

As a final step, we focused on the collective response of animals on pathogenic bacteria. It has been known that pathogenic bacteria *Pseudomonas aeruginosa* (PA14) trigger immune response

and avoidance behavior in *C. elegans* (*Reddy et al., 2011*; *Zhang et al., 2005*). In order to analyze the collective dynamics, we first measured the V(O) curves of both N2 and npr-1 strains on the lawn formed by PA14. The responses of these strains are very different from the behaviors observed on regular OP50; N2 animals are always active on the lawn, on the other hand npr-1 shows a narrowed slowing down window between 3% and 5% oxygen (*Figure 6a,c*). Further, we tested their swarming behaviors in a dense population. As expected from the V(O) curves N2 does not swarm but npr-1 shows accumulation around the lawn (*Figure 6—video 1* and *Figure 6—video 2*). Long-time lapse imaging experiments showed that the consumption of pathogenic bacteria is extremely slow which takes more than a day to finish the lawn completely.

The striking feature we observed in these experiments is the accumulation of N2 worms around the edge of the lawn (*Figure 6—video 1*). *Figure 6b* shows these clusters. The most parsimonious hypothesis explaining this observation is that bacteria can diffuse from the biofilm lawn and form a thin layer of isolated bacteria which is sufficient to change the oxygen concentration in the liquid layer on NGM surface. Previous studies showed the instability of the edge of bacterial biofilms (*Yaman et al., 2019*; *Semmler et al., 1999*). Thus, pathogenic PA14 show substantial structural differences between biofilms and edges. The bacteria in the biofilm are tightly packed and attached to the extracellular polymeric substances (EPS) matrix, however around the edge they are motile, and they can disperse using their fimbriae. Further we used GFP-labeled PA14 and observed single layer finger-like structures around the lawn (*Figure 6—figure supplement 1*). The width of this layer is about 1.6 mm for 3 days old PA14 bacterial lawn.

In contrast to N2, npr-1 animals form a large aggregate (*Figure 6d*). When we randomize the pattern and set the ambient oxygen to a higher level, we also observed transient clustering on the lawn (*Figure 6—video 2*). This observation is consistent with our diffusion concept. Animals can initially concentrate the isolated bacteria available on the surface but eventually they cannot get enough bacteria from the biofilm to sustain the aggregation.

Finally, in order to gain more information about the differences between OP50 and PA14, we imaged the dynamics of the bacteria around a single worm during locomotion. We observed that on the OP50 lawn, worms could concentrate the bacteria (*Video 11*). In contrast, on the PA14 biofilm animals only make a trench but do not concentrate the bacteria (*Figure 6—video 3*). However, around the edge of the PA14 biofilm animals can concentrate the isolated bacteria (*Figure 6—video 4*). Altogether these observations suggest that biofilm structure and diffusibility of the bacteria play essential roles during aggregation and pattern formation.

## Discussion

The dynamics of pattern formation in biological systems depend on many intricately related factors. The theory of pattern formation in active particles provides a powerful framework to explain the complex interactions between these factors (*Cates et al., 2010*; *Liebchen and Löwen, 2018*). Using Keller-Segel model (*Keller and Segel, 1970*; *Wang, 2010*) and motility-induced phase separation principles (*Liu et al., 2016*; *Cates and Tailleur, 2015*), this study throws light upon new physical and

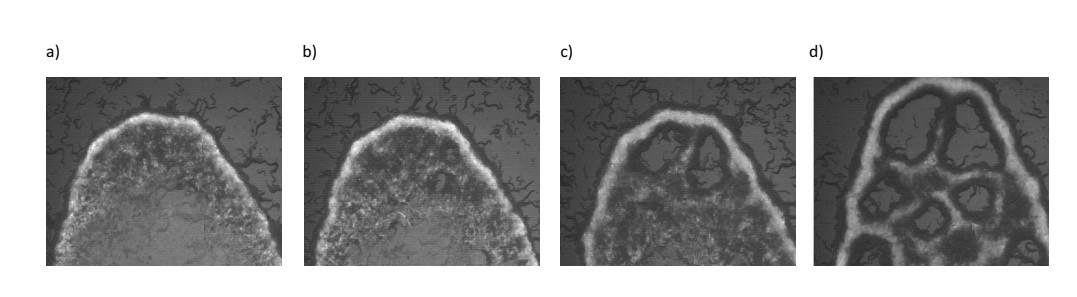

**Figure 5.** Emergence of periodic holes on biofilms. (a–d) Snapshots of aggregating N2 animals on GFP labeled *B. subtilis* biofilm. Limited diffusibility of the chaining bacteria allows local bacterial depletion and oxygen penetration. An increase in the local oxygen concentration results in periodic hole structures by triggering the animals' motility.

The online version of this article includes the following figure supplement(s) for figure 5:

**Figure supplement 1.** Time evolution of holes in a swarm body.

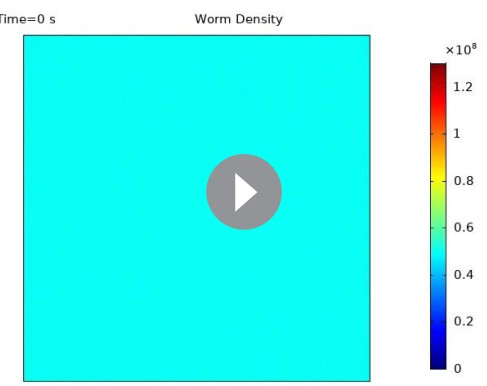

**Video 8.** Sample simulation results of representing the coarsening process.
https://elifesciences.org/articles/52781#video8

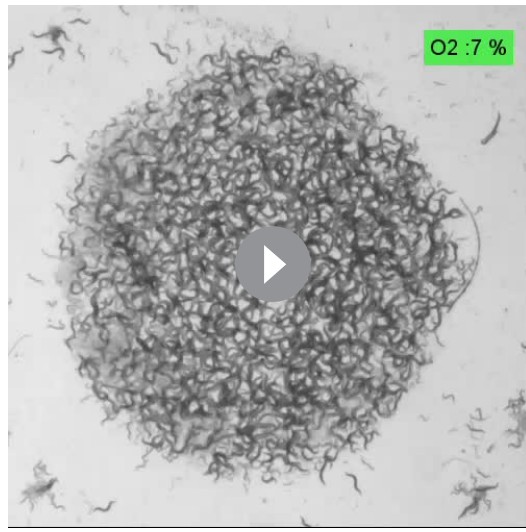

**Video 9.** Typical response of *npr-1* to changing ambient oxygen. Bacterial depletion promotes coarsening and the emergence of swarming. This video is associated with *Figure 4*.
https://elifesciences.org/articles/52781#video9

biological insights of this complex dynamics. We revealed four essential factors; First, hydrodynamic interactions between worms initiate the process of bacterial accumulation within the aggregates, which is the first, and a critical step in the pattern formation. Second, the oxygen-dependent motility of the animals controls the competition between aerotaxis and animal dispersion. This competition links the neuronal sensitivity to the collective response of the animals. Third, the population density can compensate for the neuronal sensitivity to convert the behavior of solitary animals to aggregation. Further, a gradient profile is formed across the aggregate due to the consumption of bacteria, which leads to the initiation of forward motility and swarming behavior. Finally, we observe that bacterial diffusibility and biofilm formation of the lawn strongly alter the collective dynamics of animals. Our results show that collective dynamics and pattern formation of *C. elegans* is particularly driven by slowing down response of animals. Our work also raises the questions of how the activities of sensory neurons coordinate motility and slowing down response of worms. Particularly how head and tail neurons separately sense the oxygen levels and synchronize forward and backward locomotion. Moreover, the relation between slowing down response and pathogenic avoidance behavior should be investigated. Future experiments will help identify the details of the circuit dynamics controlling these behaviors. Altogether, experimental results and mathematical models of this study will shed light on understanding the complex dynamics of biological systems.

## Materials and methods

### Derivation of the mathematical model

The motion of worms is described as a random walk with oxygen-dependent speed of worms denoted by $V(O)$. Worms can change their direction at a rate of $\tau$. The continuity equation reads as:

$$\frac{\partial W}{\partial t} = -\nabla \mathbf{J} \tag{3}$$

where $W$ is the density of worms and $\mathbf{J}$ is the worm flux and given by the following expression:

$$\mathbf{J} = -\frac{V(O)}{n\tau}\nabla(V(O)W) \tag{4}$$

n is the dimensionality and equals to 2 in our 2-D model. Substituting *Equation 4* into *Equation 3* leads to:

$$\frac{\partial W}{\partial t} = -\nabla \left[ -\frac{V(O)}{2\tau} \nabla (V(O)W) \right] \tag{5}$$

$$= \frac{1}{2\tau} \nabla \left[ V^2 \nabla W + VW \nabla V \right] \tag{6}$$

$$= \frac{1}{2\tau} \nabla \left[ V^2 \nabla W + VW \frac{\partial V}{\partial O} \nabla O \right] \tag{7}$$

$$= \nabla [D_W \nabla W] + \nabla [\beta W \nabla O] \tag{8}$$

In the equation above, $D_W = \frac{V^2}{2\tau}$ is worm diffusivity and $\beta = \frac{V}{2\tau} \frac{\partial V}{\partial O}$ is the aerotactic coupling coefficient. The oxygen kinematics can be controlled by three contributing factors. First, oxygen diffuses

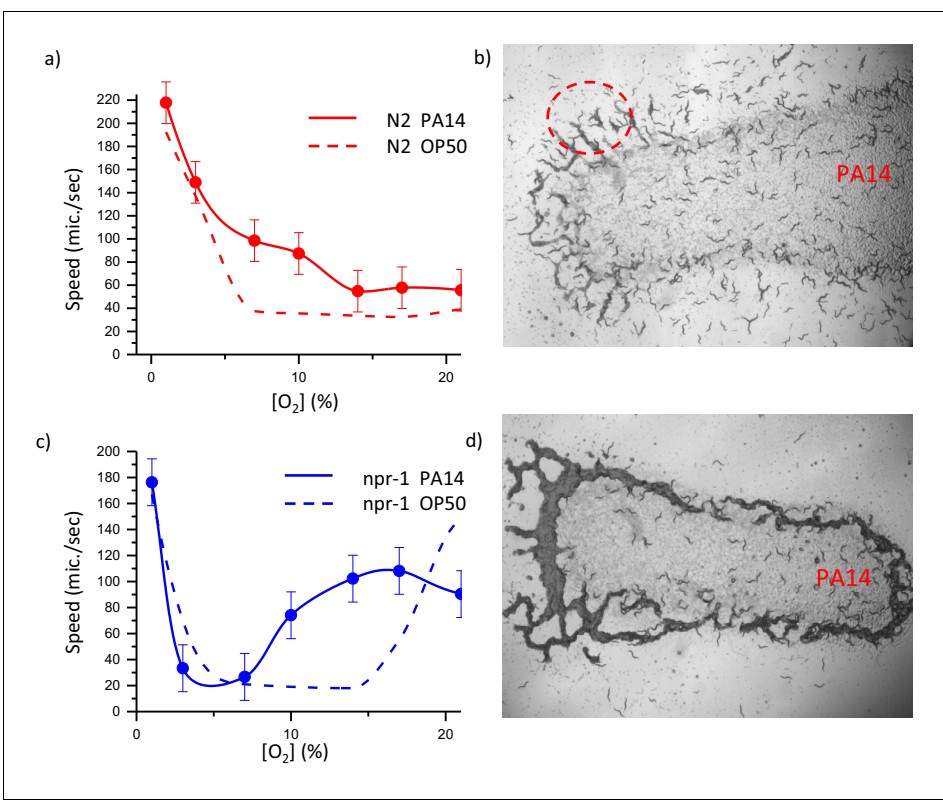

**Figure 6.** Dynamics of collective response of both N2 and npr- one on pathogenic bacteria *Pseudomonas aeruginosa* (PA14). Oxygen-dependent motility of the laboratory strain N2 (a) and (c) npr-1 on PA14. Error bar indicates the s.e.m. = ± 18 μm/s. Snapshots of distribution high-density animals N2 (b) and np1-1 (d) on PA14. For each experiment N = 25 individual animals were imaged.
The online version of this article includes the following video and figure supplement(s) for figure 6:

**Figure supplement 1.** Unstable edge of a bacterial biofilm.
**Figure 6—video 1.** Collective response of N2 animals on a biofilm-forming patghogenic *PA14* bacterial lawn.
https://elifesciences.org/articles/52781#fig6video1
**Figure 6—video 2.** Collective response of npr-1 animals on a biofilm-forming patghogenic *PA14* bacterial lawn.
https://elifesciences.org/articles/52781#fig6video2
**Figure 6—video 3.** Dynamics of bacteria around a single worm moving on the GFP-labeled PA14 biofilm.
https://elifesciences.org/articles/52781#fig6video3
**Figure 6—video 4.** Dynamics of bacteria around a single worm moving on the edge of PA14 biofilm.
https://elifesciences.org/articles/52781#fig6video4

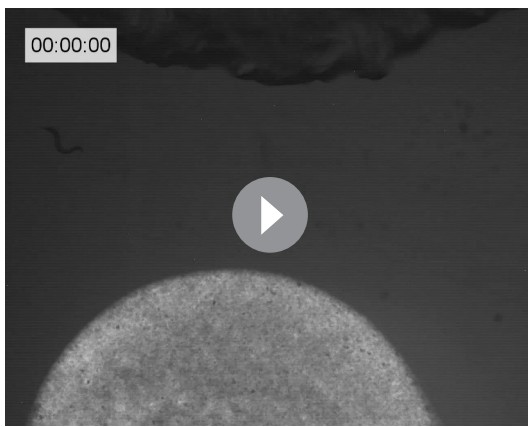

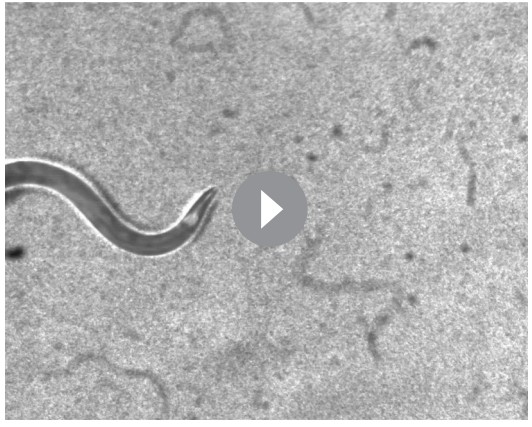

**Video 10.** Accumulation of N2 animals on a biofilm-forming *B. Subtilis* bacterial lawn. GFP-labeled bacteria show the concentrated and depleted regions. Depleted regions converge to periodic open holes. This video is associated with **Figure 5**.
https://elifesciences.org/articles/52781#video10

**Video 11.** Dynamics of bacteria around a single worm on a GFP-labeled OP50. Bacteria can diffuse and the worm accumulates the bacteria around the trench left behind.
https://elifesciences.org/articles/52781#video11

through the two-dimensional wet agar surface. The second is the penetration of the ambient oxygen from the air to the aggregate. The last contribution is the consumption of oxygen by worms and bacteria. Since the local density of worms and bacteria are linearly correlated as shown in Supplementary **Figure 3b**, the oxygen consumption can be described with the worm density. Hence, the following equation accounts for the oxygen dynamics on an agar surface:

$$\frac{\partial O}{\partial t} = D_O \nabla^2 O + f(O_{am} - O) - k_c W \tag{9}$$

Here, $D_O$ is the diffusion coefficient of the oxygen on the surface, $f$ is the penetration rate of the oxygen from the air to the water, $O_{am}$ is the ambient oxygen level and $k_c$ is the rate of oxygen consumption by worms and bacteria. We ignore the distribution of the bacteria, since it is linearly correlated with the local worm density.

## Linear stability analysis

Our equations are:

$$\frac{\partial W}{\partial t} = \nabla[D_W \nabla W] + \nabla[\beta W \nabla O] \tag{10}$$

$$\frac{\partial O}{\partial t} = D_O \nabla^2 O - k_c W + f(O_{am} - O) \tag{11}$$

At the homogeneous equilibrium state, where $W$ and $O$ are constant over time and space, we have the following solution:

$$k_c W = f(O_{am} - O) \tag{12}$$

We denote the equilibrium state as:

$$(W_{eq}, O_{eq}) = \left(W_{eq}, O_{am} - \frac{k_c}{f} W_{eq}\right) \tag{13}$$

Then we introduce small perturbations into this homogeneous equilibrium condition as follows:

$$W(x,t) = W_{eq} + \Delta W(x,t) \tag{14}$$

$$O(x,t) = O_{eq} + \Delta O(x,t) \tag{15}$$

By applying these variable replacements to *Equation 10* and *Equation 11*, we obtain

$$\frac{\partial \Delta W}{\partial t} = \frac{\partial}{\partial x}\left[D_W \frac{\partial}{\partial x}(W_{eq} + \Delta W)\right] + \frac{\partial}{\partial x}\left[\beta(W_{eq} + \Delta W)\frac{\partial}{\partial x}\left(O_{am} - \frac{k_c}{f}W_{eq} + \Delta O\right)\right] \tag{16}$$

$$= \frac{\partial}{\partial x}\left[D_W \frac{\partial}{\partial x}(\Delta W)\right] + \frac{\partial}{\partial x}\left[\beta(W_{eq} + \Delta W)\frac{\partial}{\partial x}(\Delta O)\right] \tag{17}$$

$$= \frac{\partial D_W}{\partial x}\frac{\partial \Delta W}{\partial x} + D_W \frac{\partial^2 \Delta W}{\partial x^2} + \frac{\partial \beta}{\partial x}W_{eq}\frac{\partial \Delta O}{\partial x} + \beta W_{eq}\frac{\partial^2 \Delta O}{\partial x^2} \tag{18}$$

$$\frac{\partial \Delta O}{\partial t} = D_O \frac{\partial^2}{\partial x^2}\left(O_{am} - \frac{k_c}{f}W_{eq} + \Delta O\right) - k_c(W_{eq} + \Delta W) + f\left(O_{am} - \left(O_{am} - \frac{k_c}{f}W_{eq} + \Delta O\right)\right) \tag{19}$$

$$= D_o \frac{\partial^2 \Delta O}{\partial x^2} - k_c \Delta W - f\Delta O \tag{20}$$

Second-order terms in perturbations are neglected. We assume:

$$\Delta W(x,t) = \Delta W(t)\sin kx \tag{21}$$

$$\Delta O(x,t) = \Delta O(t)\sin kx \tag{22}$$

By doing so, we have decoupled spatial structure and temporal dynamics. By definition, $D_W$ and $\beta$ are:

$$D_W = \frac{V^2}{2\tau} \tag{23}$$

$$\beta = \frac{V}{2\tau}\frac{\partial V}{\partial O} \tag{24}$$

Therefore, the partial derivatives of diffusivity and aerotactic coupling coefficient with respect to $x$ are:

$$\frac{\partial D_W}{\partial x} = \frac{1}{2\tau}2V\frac{\partial V}{\partial x} \tag{25}$$

$$= \frac{V}{\tau}\frac{\partial V}{\partial O}\frac{\partial O}{\partial x} \tag{26}$$

$$\frac{\partial \beta}{\partial x} = \frac{1}{2\tau}\left(\frac{\partial V}{\partial O}\frac{\partial V}{\partial x} + V\frac{\partial}{\partial x}\left(\frac{\partial V}{\partial O}\right)\right) \tag{27}$$

$$= \frac{1}{2\tau}\left(\left(\frac{\partial V}{\partial O}\right)^2\frac{\partial O}{\partial x} + V\frac{\partial^2 V}{\partial O^2}\frac{\partial O}{\partial x}\right) \tag{28}$$

In terms of perturbations around the equilibrium, the equation (12) becomes:

$$\frac{\partial D_W}{\partial x} = \frac{V}{\tau}\frac{\partial V}{\partial O}\frac{\partial \Delta O}{\partial x} \tag{29}$$

$$\frac{\partial \beta}{\partial x} = \frac{1}{2\tau}\left(\left(\frac{\partial V}{\partial O}\right)^2\frac{\partial \Delta O}{\partial x} + V\frac{\partial^2 V}{\partial O^2}\frac{\partial \Delta O}{\partial x}\right) \tag{30}$$

Since each term in the expressions of diffusivity and aerotactic coupling coefficient parameter is first order in small perturbations, their multiplication with another first-order term will be negligible. Therefore, we neglect all higher order terms in *Equation 18*. Finally, these equations become:

$$\sin kx\frac{\partial \Delta W}{\partial t} = -D_W k^2\sin kx\Delta W - \beta W_{eq}k^2\sin kx\Delta O \tag{31}$$

$$\frac{\partial \Delta W}{\partial t} = -D_W k^2\Delta W - \beta W_{eq}k^2\Delta O \tag{32}$$

$$\sin kx\frac{\partial \Delta O}{\partial t} = -D_O k^2\sin kx\Delta O - k_c\sin kx\Delta W - f\sin kx\Delta O \tag{33}$$

$$\frac{\partial \Delta O}{\partial t} = -D_O k^2\Delta O - k_c\Delta W - f\Delta O \tag{34}$$

Using a linear algebra notation:

$$\frac{\partial}{\partial t}\begin{pmatrix}\Delta W \\ \Delta O\end{pmatrix} = \begin{pmatrix}-D_W k^2 & -\beta W_{eq}k^2 \\ -k & -D_O k^2 - f\end{pmatrix}\begin{pmatrix}\Delta W \\ \Delta O\end{pmatrix} \tag{35}$$

We need to find the eigenvalues of the matrix at *Equation 35* and check the signs of their real parts.

$$\begin{vmatrix}-D_W k^2 - \lambda & -\beta W_{eq}k^2 \\ -k_c & -D_O k^2 - f - \lambda\end{vmatrix} = 0 \tag{36}$$

$$\left(-D_W k^2 - \lambda\right)\left(-D_O k^2 - f - \lambda\right) - \beta W_{eq}k^2 k_c = 0 \tag{37}$$

$$\lambda^2 + \left(D_W k^2 + D_O k^2 + f\right)\lambda + D_W k^2\left(D_O k^2 + f\right) - \beta W_{eq}k^2 k_c = 0 \tag{38}$$

$$\lambda = \frac{1}{2}\left(-\left(D_W k^2 + D_O k^2 + f\right) \pm \sqrt{\left(D_W k^2 + D_O k^2 + f\right)^2 - 4\left(D_W k^2(D_O k^2 + f) - \beta W_{eq}k^2 k_c\right)}\right) \tag{39}$$

Eigenvalue must be positive for an unstable condition. Since all the terms before ± are negative, the square root term must be positive and sufficiently large in order to make the real part of the eigenvalue positive. Therefore, the condition for a positive real part is:

$$D_W k^2 + D_O k^2 + f < \sqrt{\left(D_W k^2 + D_O k^2 + f\right)^2 - 4\left(D_W k^2(D_O k^2 + f) - \beta W_{eq}k^2 k_c\right)} \tag{40}$$

$$\left(D_W k^2 + D_O k^2 + f\right)^2 < \left(D_W k^2 + D_O k^2 + f\right)^2 - 4\left(D_W k^2(D_O k^2 + f) - \beta W_{eq}k^2 k_c\right) \tag{41}$$

$$D_W k^2\left(D_O k^2 + f\right) < \beta W_{eq}k^2 k_c \tag{42}$$

If this inequality holds, the homogenous equilibrium state is unstable. By solving the inequality for $k^2$, we find:

$$\frac{W_{eq}\beta k_c}{D_W} - f > D_O k^2 \tag{43}$$

Since $k^2$ and $D_O$ are positive, we get the following result for instability:

$$W_{eq}\beta k_c > f D_W \tag{44}$$

## Numerical solution

We used numerical methods to solve the coupled partial differential equations. To solve the equations in time and space, we used COMSOL Multiphysics, finite element methods and periodic boundary conditions in space. The coupled partial differential equations are implemented by using general form coupled differential equations:

$$\frac{\partial W}{\partial t} = \nabla\left[\frac{V^2}{2\tau}\nabla W + \frac{V}{2\tau}\frac{\partial V}{\partial O}W\nabla O\right] \tag{45}$$

$$\frac{\partial O}{\partial t} = D_O\nabla^2 O + f(O_{am} - O) - k_c W \tag{46}$$

Here, we replaced the dispersion and aerotactic coupling coefficients with their definition. In our simulations, we used $V$ as a parabolic function of $O$ which has the following expression:

$$V = aO^2 + bO + c \tag{47}$$

At the beginning of the simulations, worms are uniformly distributed with additional noise. The parameters used in the simulations are given below. f and $k_c$ are chosen to acquire the width of experimentally observed domain boundaries and time dynamics. $D_O, D_W$, $\beta$, $W_{eq}, O_{am}$ and velocity parameters are experimentally defined. Realistic worm densities and diffusion were measured by imaging the worms cluster by setting ambient oxygen to 0.

Parameters of velocity (m/s): $a = 1.89E - 2$, $b = -3.98E - 3$, $c = 2.25E - 4$
Tumbling rate: $\tau = 0.5$ 1/s
Oxygen diffusion coefficient: $D_O = 2E - 5 \; cm^2/sec$
Oxygen penetration rate: $f = 0.65 \; 1/sec$
Oxygen consumption rate by worms and bacteria: $k_c = 7.3 \; E - 10 \; 1/sec$
Ambient oxygen level $O_{am}$ ranges between 0 and 0.21
Worm density W, in the uniform stage is around ~ 1–90 worms/mm$^2$
Width and height of the simulation window; $L = 2cm$

## *C. elegans* strains

Strains were grown and maintained under standard conditions unless indicated otherwise. All the strains were obtained from Caenorhabditis Genetics Center (CGC).

Strain List: *npr1* (DA609), *npr1* (CX4148), *gcy35* (AX1295), *cat2* (CB1112), *tph1* (MT15434), *bas1* (MT7988), *bas1 and cat4* (MT7983), *tax2* (PR694), *tax4* (PR678), *tax2 and tax4* (BR5514), *nsy1* (AWC off killed-CX4998), *eat2* (DA465), *mec4* (CB1611), *mec10* (CB1515), *osm9* (VC1262), *trpa1* (RB1052), *dig1* (MT2840), *trp4* (VC1141), *lite1* (ce314), *daf19 and daf12* (JT6924), *npr-1(ad609) X*.

## Swarming protocol

Nematode growth media (NGM) plates having a diameter of 9 cm were used for maintaining the worms. NGM plates were seeded with 1 ml of OP50 culture. After the worms consumed the bacteria, three NGM plates were washed with $1 \times$ M9 buffer and the wash was centrifuged twice at 2000 rpm for 30 s. Centrifugation was repeated (up to six times) until a clear supernatant was obtained. Since multiple centrifugation cycles might affect the activity of worms, the cycles subsequent to the first two cycles were carried out for 10 s at 2000 rpm. After cleaning the worms, a 150 µl worm droplet was put on a 6 cm plate seeded with 100 µl ($0.5 \times 1$ cm) of OP50 culture. The optimum swarming pattern was observed in 6-day-old plates and hence 6-day-old plates were used in all further experiments unless indicated otherwise. For NGM plates, M9 buffer preparation and worm synchronization were done based on the standard protocols given in the wormbook.

## Oxygen control and measurements

A plexiglass chamber was used to control the ambient [O$_2$] (*Figure 2—figure supplement 4*). A 50 sccm mixture of O$_2$ and N$_2$ was used and flow rates were controlled by a flow controller. [O$_2$] was

measured by using a normal oxygen sensor (PreSens, Microx TX3). During [O2] measurements the sensor probe should be inserted entirely into swarm fluid in order to read valid signals. The fiber optic sensor has a polymer coating and O2 can easily diffuse from the open air-exposing regions and can quench the fluorescence signal. During fiber optic removal, capillary meniscus around the fiber also affects the reading, we only read the ambient signal [O2] when the meniscus breaks.

At the beginning of the swarming experiments, the oxygen levels in the chamber were adjusted to 21%. After swarming was first observed, oxygen levels were sequentially decreased to 10%, 7%, 3%, and 1% at an interval of ten minutes. Then, oxygen levels were increased in reverse order. The experiments were recorded using Thorlabs DCC1545M CMOS camera with a Navitar 7000 TV zoom lens.

## Acknowledgements

This work was supported by an EMBO installation grant (IG 3275); The Science Academy, Turkey, (BAGEP) young investigator award and TUBITAK (project no115F072 and 115S666). We thank Tevfik Can Yuce for computational support. We thank Mario de Bono and his group members for discussions and suggestions about the initial observation of the swarming response. We thank Prof FM Ausubel for providing GFP labeled PA14 strain. We also thank H Kavakli, M Iskin, F Balci, P A Ramey for the comments and critical reading of the manuscript.

## Additional information

### Funding

| Funder | Grant reference number | Author |
| --- | --- | --- |
| European Molecular Biology Organization | IG 3275 | Askin Kocabas |

The funders had no role in study design, data collection and interpretation, or the decision to submit the work for publication.

### Author contributions

Esin Demir, Data curation, Formal analysis, Observed the swarm response, Performed swarning experiments and mutant screens, Wrote the manuscript; Y Ilker Yaman, Conceptualization, Data curation, Software, Formal analysis, Investigation, Methodology, Performed the pattern formation experiments, designed the imaging systems, developed the mathematical models, performed the simulations and carried out image processing analysis of experiments, Wrote the manuscript; Mustafa Basaran, Conceptualization, Data curation, Software, Formal analysis, Investigation, Methodology, Performed biofilm experiments, Developed the COMSOL implementation, Wrote the manuscript; Askin Kocabas, Conceptualization, Resources, Data curation, Software, Formal analysis, Supervision, Funding acquisition, Validation, Investigation, Visualization, Methodology, Project administration, Performed the pattern formation experiments, designed the imaging systems, developed the mathematical models, performed the simulations and carried out image processing analysis of experiments, Performed biofilm experiments, Wrote the manuscript

### Author ORCIDs

Y Ilker Yaman https://orcid.org/0000-0003-4094-616X
Mustafa Basaran https://orcid.org/0000-0002-1895-254X
Askin Kocabas https://orcid.org/0000-0002-6930-1202

### Decision letter and Author response

Decision letter https://doi.org/10.7554/eLife.52781.sa1
Author response https://doi.org/10.7554/eLife.52781.sa2

## Additional files

### Supplementary files

- Transparent reporting form

### Data availability

All data generated or analyzed during this study are included in the manuscript and supporting files.

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
