## [Decision Letter]

**Acceptance summary:**

The manuscript describes the rich phenomenology of swarming behavior of *C. elegans* worms on a bacterial lawn, forming Turing-like patterns under conditions of low density and oxygen. The study integrates a comprehensive array of experiments with a simple theory, which allows the authors to show that a combination of ambient [O_2_] and bacterial structure and diffusibility play ‎critical roles during this collective swarming behavior.

**Decision letter after peer review:**

Thank you for submitting your article "Dynamics of pattern formation and emergence of swarming in *C. elegans*" for consideration by *eLife*. Your article has been reviewed by Aleksandra Walczak as the Senior Editor, a Reviewing Editor, and three reviewers. The following individuals involved in review of your submission have agreed to reveal their identity: Alon Zaslaver (Reviewer #1), Vishwesha Guttal (Reviewer #2); Davide Marenduzzo (Reviewer #3).

The reviewers have discussed the reviews with one another and the Reviewing Editor has drafted this decision to help you prepare a revised submission.

All the reviewers have found the swarming behavior intriguing and interesting. However, they have raised some concerns about the interpretation of the observations. In particular, they have suggested some control experiments and additional analyses to rule out alternative explanations and probe the underlying assumptions you have made. Please address the comments in the three reviews attached below in your revision.

*Reviewer #1:*

The authors describe an interesting complex behavior of *C. elegans* worms as they collectively swarm ‎through a bacterial food lawn: Initially, solitary worms reach the edge of the bacterial lawn and start to ‎pile up. The intricate concentrate of worms in the aggregate reduces [O_2_] and as a result worms remain ‎in the aggregate. The accumulated pile of worms then gets motion once the worms consume the bacteria ‎within the aggregate. The swarm moves along the lawn and consumes all the bacteria, after which the ‎worms disperse. In sum, a combination of ambient [O_2_] and bacterial structure and diffusibility play ‎critical roles during this collective swarming behavior.‎

Parts of the data presented in this manuscript had been shown before (e.g, [O_2_] reduction within the ‎worms' aggregate, the speed of various mutants across a wide range of [O_2_], and worm's [O_2_] ‎preference; see for example: Gray… et al., 2004, and Rogers etal…, 2006.‎

Yet, the authors herein put all these data, as well as additional supporting results and including a ‎mathematical model, in a nice comprehensive framework that describes this intriguing collective ‎behavior.‎

Essential revisions:‎

1) The authors interpret that worms' aggregate makes the bacteria more ‎concentrated (Results section). The edge of the bacterial lawn is considerably thicker than the inside ‎of the lawn, especially when using several-days old lawns, as described in this study. The ‎worms therefore have much space to burrow into this thick border of bacteria and feed on them. ‎This by itself suppresses motion. Solitary worms reaching such edges also typically suppress ‎motion and remain in the lawn periphery. Once this region is consumed, the worms, which are ‎already piled up to form an aggregate, continue swarming along the bacterial lawn. It may be that there is no need to ‎concentrate the bacteria further to get the observed behavior as suggested.‎ Could the authors comment on that or maybe discuss this as an alternative explanation?

2) The assumption throughout the manuscript is that the bacteria deplete [O_2_] in the aggregate (for ‎example, Figure 1—figure supplement 2). It could, however, be that the viscous solution of ‎bacteria is poorly permeable to ambient O_2_. The authors can easily check if it is actually ‎bacterial respiration that consumes all O_2_ by measuring O_2_ level in an aggregate consisted of dead ‎bacteria (e.g. heat killed bacteria). This will help to support this claim.

‎3)‎ Figure 2B: not understood at all. Is the average speed of individual worms shown in the graph? ‎This is not indicated in the legend nor in the Materials and methods section. The green data points which indicate off-‎food, are they for the N2 or the npr-1 strain? Also, the legend says error bars are SEM, but no ‎error bars appear in the graph. Does this mean that the indicated 18% SEM hold for all data ‎points?‎

‎4)‎ Figure 3B: legend indicates of error bars but these are not presented. Also, does it show a ‎representative measure or an average of several measurements?‎

‎5)‎ Results section: It is noted that glb-5 can form small clusters even at low densities, but no value is ‎given for these densities. How is this value compared to N2 and npr-1 worms? The change in ‎cluster shown in Figure 3C regarding glb-5 different, is it statistically significant different from ‎what is found in npr-1 and N2 worms? ‎

‎6)‎ Substantial grammar, punctuation, and style revisions are required. Some parts of the manuscript ‎were very difficult to understand.‎

*Reviewer #2:*

I really enjoyed reading the article on dynamic pattern formation and emergence of swarming in *C. elegans*. Authors conduct a remarkably comprehensive array of experiments which are exceptionally well matched with a simple theory that the authors develop/modify based on existing models of the literature. Authors show that *C. elegans* form Turing-like patterns under conditions of low density and oxygen conditions. Such integrative studies are rare, and hence authors deserve much appreciation for the rigour of the work and very fascinating results they have demonstrated. I am not an expert on *C. elegans* or its mechanisms of motility or swarming. So my review is broadly based on the novelty and comprehensiveness of the study.

I do not have any major comments that I would like the authors to address. However, I am keen to know whether the local positive feedback + long-range inhibition (also called scale-dependent feedback), which is thought to be a fairly universal principle behind such regular Turing-like patterns, is also true in this system. I would appreciate if the authors comment on this aspect. One reference that may be useful in this context is the paper by Rietkerk and van de Koppel, 2008. Although this is written from the perspective of ecosystems and pattern formation, the broader principle of how scale-dependent feedbacks shape patterns will be relevant.

I would also like to see a short discussion on potential alternative mechanisms that may explain these patterns, and also some future directions based on this work.

*Reviewer #3:*

The authors describe the pattern formation in a system with *C. elegans* on bacterial lawns or biofilms. They argue that the patterns form because motility decreases with oxygen so that motility-induced phase separation sets up. On top of this, bacteria are consumed by worms and this sets up a swarming behaviour at sufficiently late times. I think this work shows very interesting experiments and the videos are useful to understand the very rich phenomenology.

The main concern that I think should be addressed in full before publication is with the numerical simulation of the continuum model, which is based on a combination of Keller-Segal equations and motility-induced phase separation ideas. I think the overall motivation/basis of the modelling is sound, but it is unclear to me whether more than quite a qualitative comparison can be made. An important feature concerns coarsening, or macroscopic phase separation versus microphase separation. This is briefly discussed in Figure 5 and in some of the supplementary Videos it is shown that late-time behaviours feature patterns coarsening into a single swarming pattern. But this applies to experiments only and it is related according to the authors to bacteria consumption.

In the simulations, the times at which the various snapshots in Figure 2 and Figure 3 are shown is not specified. Presumably the times are all the same but it should be mentioned. If I understand correctly, from the dispersion relation it seems to me that in the simulations, which assume no bacterial consumption, there is still coarsening rather than microphase separation and the formation of multiple stable patterns as for instance in MIPS with logistic bacterial replication (Cates et al., 2010). Can the authors clarify whether this is the case? In other words, they should discuss the fate of the simulation patterns at late times. If all patterns coarsen, then it is important to motivate the choice of the intermediate times at which snapshots are shown (as, if this were the case, the end result would be in all cases complete phase separation). Related to this, I did not have the impression that the patterns would coarsen experimentally at timescales for which bacterial consumption becomes relevant. It would be good to comment on this as well.

---

## [Author Response]

Reviewer #1:The authors describe an interesting complex behavior of *C. elegans* worms as they collectively swarm ‎through a bacterial food lawn: Initially, solitary worms reach the edge of the bacterial lawn and start to ‎pile up. The intricate concentrate of worms in the aggregate reduces [O2] and as a result worms remain ‎in the aggregate. The accumulated pile of worms then gets motion once the worms consume the bacteria ‎within the aggregate. The swarm moves along the lawn and consumes all the bacteria, after which the ‎worms disperse. In sum, a combination of ambient [O2] and bacterial structure and diffusibility play ‎critical roles during this collective swarming behavior.‎Parts of the data presented in this manuscript had been shown before (e.g, [O2] reduction within the ‎worms' aggregate, the speed of various mutants across a wide range of [O2], and worm's [O2] ‎preference; see for example: Gray et al., 2004, and Rogers etal., 2006.‎Yet, the authors herein put all these data, as well as additional supporting results and including a ‎mathematical model, in a nice comprehensive framework that describes this intriguing collective ‎behavior.‎

We thank the reviewer for a general review and emphasizing the contribution of the manuscript.

Essential revisions:‎1 The authors interpret that worms' aggregate makes the bacteria more ‎concentrated (Results section). The edge of the bacterial lawn is considerably thicker than the inside ‎of the lawn, especially when using several-days old lawns, as described in this study. The ‎worms therefore have much space to burrow into this thick border of bacteria and feed on them. ‎This by itself suppresses motion. Solitary worms reaching such edges also typically suppress ‎motion and remain in the lawn periphery. Once this region is consumed, the worms, which are ‎already piled up to form an aggregate, continue swarming along the bacterial lawn. It may be that there is no need to ‎concentrate the bacteria further to get the observed behavior as suggested.‎ Could the authors comment on that or maybe discuss this as an alternative explanation?

We would like to thank the reviewer for pointing out this interesting point. The reviewer is predicting that around thick bacterial region worms do not need to concentrate bacteria and they should not form aggregates.

We fully agree with the reviewer’s prediction, based on our model worms should not form aggregates if the bacterial layer is thick enough. Bacterial thickness is the main parameter controlling the effective oxygen level. First, we classified this scenario into three different conditions,

1)in athin bacterial layer, animals need to concentrate the bacteria to create preferred oxygen levels (7-10%). This is the condition we observe on regular NGM plates. Bacteria cannot grow efficiently and form only a thin layer (5-10 mic). The edge of the lawn is relatively thick but still thinner than the worm thickness. We verified this edge thickness using a digital microscope (Author response image 1). The other important factor here is the worm’s motion, they can easily spread and flatten the thick bacterial regions around the edge.

**Author response image 1. respfig1:** 3D topography of 10 days old OP50 bacterial lawn on regular NGM plate. The edge thickness is around 16 μm.

2) The second case is the relatively thick bacterial layer (~50 mic). If the layer is comparable with the worm thickness, they do not need to form aggregates. On a regular plate, we may rarely observe this condition locally, but again worms can easily damage these concentrated regions.

3) The last case is the thick bacterial layer (>50 mic). In this condition, bacteria cover the entire worm’s body and they can only experience very low oxygen. Thus, they should definitely escape from these regions.

We decided to test these conditions experimentally, particularly we tried to observe the last two cases. We added yeast extract to NGM plates to promote aggressive bacterial growth (OP50). We tested the aggregation response of npr-1 worms on this very thick bacterial lawn. We transferred the worms from regular NGM plate to thick bacterial lawn. As predicted, animals do not form aggregates. Figure 2—figure supplement 3 compares the aggregation responses of npr-1 worms on both plates. It is evident that worms do not form clusters. We further imaged the worms for several days and observed that worms escape from very thick bacterial regions (Video 8 dark area around the center). During this long time-laps imaging we do not observe any aggregate. After 4 -5 days they become very crowded and depleted the bacteria. After then we started seeing the aggregates.

We have added all these control experiments to the manuscript. We would like to thank the reviewer for this prediction. We think that this is a significant contribution to support our model.

2) The assumption throughout the manuscript is that the bacteria deplete [O2] in the aggregate (for ‎example, Figure 1—figure supplement 2). It could, however, be that the viscous solution of ‎bacteria is poorly permeable to ambient O2. The authors can easily check if it is actually ‎bacterial respiration that consumes all O2 by measuring O2 level in an aggregate consisted of dead ‎bacteria (e.g. heat killed bacteria). This will help to support this claim.

We thank the reviewer for this suggestion. We agree that other factors could have additional effects on oxygen kinematics. Our previous experiment (Video 7) shows that the presence of oxygen depleting chemicals such as NaSO3 could also induce aggregation response without bacteria. In order to test the effects of dead bacteria, we performed additional experiments. We used heat-killed bacterial lawn (overnight grown OP50, 1hour, 65 ^0^C). As a control experiment, an LB plate was seeded with 100 μl bacterial suspension and we did not observe any colony. We found that npr-1 worms can aggregate on a dead bacterial lawn. We also noticed that overnight grown bacterial suspension has very low oxygen levels ([O]~ 0.1%). Killing bacteria does not increase this oxygen level. This suggests that there is no difference between live and dead bacteria if we use their native solution. Therefore, we washed the dead bacteria with fresh M9 buffer and seed the plate with these dead bacteria. Again, we observed slow accumulation around the edge. The Author response image 2 shows the snapshots of these accumulated worms after 30 minutes. Accumulation dynamics are very different.

**Author response image 2. respfig2:** Aggregation behaviors of *npr-1* worms on dead bacterial lawns. Op50 bacterial suspension was heat-killed 65 0C before seeding. Bacterial lawns were prepared (**a**) with native suspension (**b**) fresh M9 washed suspension. *npr-1* worms can form aggregation on a dead bacterial lawn.

Further in order to clarify this point, we measured the permeability of dead bacterial suspension washed with fresh M9 buffer. We sandwiched the fiber optic sensor with cover glass and let the oxygen diffuse only from the open end. Before loading the dead bacterial suspension, we shook the sample to saturate oxygen. Later we set the ambient oxygen level to 0% and then 21% and measured the oxygen dissociation rate. Successively we repeated the same procedure with M9 buffer, 1X, 10x and 50 X concentrated dead bacterial suspension. Author response image 3 below shows that M9 buffer and dead bacterial suspensions have different dissociation rates depending on the bacterial density. These results suggest that suspensions with dead bacteria have low oxygen capacity. On the other hand, dead bacteria with its own overnight suspension has very low oxygen content and it is definitely enough to decrease oxygen on the surface.

**Author response image 3. respfig3:** Oxygen dissociation rate measurements of dead bacterial suspensions. Dense bacterial suspensions quickly release oxygen.

When we use M9 washed-dead bacteria as a lawn, animals form small clusters. Unfortunately, in these small clusters, measuring oxygen concentration by using a fiber optic sensor is not possible. Although we do not know the exact mechanism, we can conclude that dead bacteria can decrease the oxygen both in their native solution or in fresh M9 buffer. Furthermore, dissociation rates of oxygen in dead bacterial suspension are different, it can quickly release oxygen. This simply suggests that worms on this layer can consume oxygen very quickly. At this stage we are not able to make a strong claim about the permeability or the solubility of oxygen in these suspensions. More effort is needed to understand the exact mechanism behind the effects of dead bacteria. We also tested Sephadex (G-50) beads and observed small but notable clusters around the beads. Our experiments are not conclusive enough at this stage, so we would like to leave these results and the discussion in the review file.

We have modified the following paragraph to emphasize this point.

“We also tested animal responses on a very thick bacterial lawn which can significantly decrease the oxygen level. We found that npr-1 worms did not form clusters on the thick bacterial lawn and they avoided extremely thick regions of the lawn (Video 8). As a final control experiment we tested the aggregation responses of worms on a dead OP50 lawn. Interestingly dead bacterial suspensions have different oxygen kinematics and results in small clustering (more discussion about dead bacteria can be found in the review file). From these results we can conclude that animals effectively experience the average O_2_ levels defined by the ambient environment and the swarm liquid. Altogether, ambient oxygen levels and oxygen depletion are essential factors for aggregation.”

‎3)‎ Figure 2B: not understood at all. Is the average speed of individual worms shown in the graph? ‎This is not indicated in the legend nor in the Materials and methods section. The green data points which indicate off-‎food, are they for the N2 or the npr-1 strain? Also, the legend says error bars are SEM, but no ‎error bars appear in the graph. Does this mean that the indicated 18% SEM hold for all data ‎points?‎

We clarified this figure with additional explanations. We would like to thank the reviewer for highlighting the invisibility of error bars. We have fixed this problem in the graph. The green curve corresponds to N2 worms. Npr-1 worms also have a very similar response but not given in the figure. We also clarified this point.

Error bars show the error of measurement. In these experiments, low magnification and image processing steps define the error of the speed measurement. Particularly, this is important for slow worms we observe around 10% oxygen levels where the worms are almost stationary. We prefer indicating the measurement error. We have explained this part in the figure legend.

‎4)‎ Figure 3B: legend indicates of error bars but these are not presented. Also, does it show a ‎representative measure or an average of several measurements?‎

Figure 3B is a single measurement. Synchronization and averaging different measurements are not possible here. The error of the measurement was calculated by averaging independent oxygen measurements

‎5)‎ Results section: It is noted that glb-5 can form small clusters even at low densities, but no value is ‎given for these densities. How is this value compared to N2 and npr-1 worms? The change in ‎cluster shown in Figure 3C regarding glb-5 different, is it statistically significant different from ‎what is found in npr-1 and N2 worms? ‎

We agree that this part of the work is not well supported, and the quantitative comparison is not possible at this stage. We have decided to remove all the experiments about glb-5.

‎6)‎ Substantial grammar, punctuation, and style revisions are required. Some parts of the manuscript ‎were very difficult to understand.‎

We have edited some parts of the manuscript to readability of the text, and we received comments particularly from colleagues with different backgrounds to improve the flow of the paper.

Reviewer #2:I really enjoyed reading the article on dynamic pattern formation and emergence of swarming in *C. elegans*. Authors conduct a remarkably comprehensive array of experiments which are exceptionally well matched with a simple theory that the authors develop/modify based on existing models of the literature. Authors show that *C. elegans* form Turing-like patterns under conditions of low density and oxygen conditions. Such integrative studies are rare, and hence authors deserve much appreciation for the rigour of the work and very fascinating results they have demonstrated. I am not an expert on *C. elegans* or its mechanisms of motility or swarming. So my review is broadly based on the novelty and comprehensiveness of the study.

We would like to thank the reviewer for motivating comments and a general review of the manuscript.

I do not have any major comments that I would like the authors to address. However, I am keen to know whether the local positive feedback + long-range inhibition (also called scale-dependent feedback), which is thought to be a fairly universal principle behind such regular Turing-like patterns, is also true in this system. I would appreciate if the authors comment on this aspect. One reference that may be useful in this context is the paper by Rietkerk and van de Koppel, 2008. Although this is written from the perspective of ecosystems and pattern formation, the broader principle of how scale-dependent feedbacks shape patterns will be relevant.I would also like to see a short discussion on potential alternative mechanisms that may explain these patterns, and also some future directions based on this work.

We thank the reviewer for bringing up this point. Following the reviewers’ suggestions, we checked the literature particularly about ecological pattern forming examples. We noticed that vegetation patterning has strong similarities with our system. We linked these seemingly unrelated systems by comparing their feedbacks and variables. Water limited dryland vegetation can form complex patterns. The models representing their dynamics are based on three variables, biomass density, water content and surface water thickness. Interestingly the first similarity appears between these variables. Our model has worm density, oxygen content and bacterial layer. Like bacterial diffusion in our system vegetation patterns strongly depend on the flow properties of the surface water layer. On the other hand, water infiltration by biomass acts as local feedback and water flow sets long-range feedback on the system. In our case, oxygen consumption is local and oxygen penetration and diffusion can be considered as competing for long-range feedback on the system. From the activator-inhibitor perspective there is no one to one similarity. This is because, in our model, worm diffusion is not constant, and it includes oxygen taxis component which depends on animals’ motility. We can only define an effective diffusion coefficient. For low oxygen regime we can claim that worm density activates lack of oxygen and lack of oxygen inhibits worm density. Here, worm density can be considered as an activator and the lack of oxygen is the inhibitor. However, for higher oxygen regime, worms have negative effective diffusion due to oxygen taxis. In this regime we cannot define inhibitory feedbacks. We have added this additional information to emphasize the similarities and differences.

Besides strong similarities between these systems, the main difference is the active matter nature of the worms. This is completely different from the dynamics of passive particle models. Although, we are not able to fully implement all the properties, motility-based diffusion can provide some of them. Comments of the last reviewer are all about these differences. As a future direction we are proposing to use optogenetic perturbation approach to fully extract the internal dynamics. These experiments will allow us to control not only the neural activity but also the speed of the worms and will reveal the activity-based contributions leading to pattern formation. We modified the manuscript as given below and shortly added this discussion.

“To gain a more quantitative representation of pattern formation, we developed a mathematical model. Many different models have been used to describe the dynamics of pattern formation. Particularly dryland vegetation models share similarities in terms of dynamic variables and feedbacks. However, in our system we have to implement the activity of the worms based on oxygen concentrations. To do so we followed the notation and the framework developed for active chemotactic particles.”

Reviewer #3:The authors describe the pattern formation in a system with *C. elegans* on bacterial lawns or biofilms. They argue that the patterns form because motility decreases with oxygen so that motility-induced phase separation sets up. On top of this, bacteria are consumed by worms and this sets up a swarming behaviour at sufficiently late times. I think this work shows very interesting experiments and the videos are useful to understand the very rich phenomenology.

We would like to thank the reviewer for his comments and a general assessment of the work.

The main concern that I think should be addressed in full before publication is with the numerical simulation of the continuum model, which is based on a combination of Keller-Segal equations and motility-induced phase separation ideas. I think the overall motivation/basis of the modelling is sound, but it is unclear to me whether more than quite a qualitative comparison can be made.

We would like to thank the reviewer for raising this point. Many of the quantities used in the simulation were experimentally verified. Dw, DO, V(O), β(Ο) worm densities etc. We are actually very close to quantitative comparison. The only exemption is the kc (rate of oxygen consumption) and f (the penetration rate of oxygen). These two variables are still arbitrary. We have chosen them to match the experimentally observed time dynamics and also the penetration depth around the domain boundary of worm clusters. The figure below shows the sample simulation results of the worm density and oxygen profiles. Using these quantities, the penetration depth is about ~200-400 µm. This penetration is around a few worm thicknesses which is reasonable compared to our experimental observation. We tried to extract these variables experimentally however, our fiber optic sensor has a very poor spatial resolution. The smallest sensor tip is around 100 μm. Currently, it is not possible to further verify these two quantities experimentally. Very recently we have found an alternative approach to measure oxygen profile precisely in micron-scale. This method depends on the photoconversion of GFP to RFP under a low oxygen regime (in vivo oxygen imaging using green fluorescent protein, Takahashi, 2006). This interesting method is very promising but depends on photobleaching upon strong light exposure. We are still having some difficulties with the implementation of this approach. Because, strong light exposure triggers avoidance of the worms which quickly disrupts the measurements. We added Figure 3—figure supplement 4 and explain how we set the physical quantities in our simulation to clarify this point.

An important feature concerns coarsening, or macroscopic phase separation versus microphase separation. This is briefly discussed in Figure 5 and in some of the supplementary Videos it is shown that late-time behaviours feature patterns coarsening into a single swarming pattern. But this applies to experiments only and it is related according to the authors to bacteria consumption.

We observe both coarsening and also the convergence of clusters to the swarming phase. We thank the reviewer for highlighting this confusion. We modified the text and use coarsening to define the initial growth of the structures.

In the simulations, the times at which the various snapshots in Figure 2 and Figure 3 are shown is not specified. Presumably the times are all the same but it should be mentioned.

We have updated the simulation results; all the images are taken at time t=10 min after the randomization of the pattern. This time window is experimentally reasonable because we do not observe significant bacterial consumption within this duration.

If I understand correctly, from the dispersion relation it seems to me that in the simulations, which assume no bacterial consumption, there is still coarsening rather than microphase separation and the formation of multiple stable patterns as for instance in MIPS with logistic bacterial replication (Cates et al., 2010). Can the authors clarify whether this is the case? In other words, they should discuss the fate of the simulation patterns at late times. If all patterns coarsen, then it is important to motivate the choice of the intermediate times at which snapshots are shown (as, if this were the case, the end result would be in all cases complete phase separation). Related to this, I did not have the impression that the patterns would coarsen experimentally at timescales for which bacterial consumption becomes relevant. It would be good to comment on this as well.

The referee is correct that patterns in our simulation are coarsening. In order to clarify this point we calculated the characteristic domain size of simulated patterns and compared them with the experimental measurements. The figure below shows that the exponent of the curve is close to the universal value of 1/3. On the other hand, experimental results reach saturation around L~ 2mm. After this time bacterial depletion dominates the dynamics and all the clusters dissociate. Here we are speculating that this saturation is defined by the bacterial diffusion and consumption within the worm clusters. GFP labeled bacterial gradient provides a comparable length scale. This mechanism resembles the arrested phase separation mechanism due to bacterial death (Cates et al., 2010). Our simulation unfortunately does not have bacterial consumption dynamics. This made the simulation unstable. We were not able to find realistic parameters forming patterns yet. We added Figure 4—figure supplement 3 to the manuscript and modified the text to clarify this point.

“The other interesting feature of the pattern formation is the coarsening event. We sought to know how the shape of the patterns evolves in time at a fixed oxygen level. Our time-lapse imaging and simulation results validate the coarsening of charanteristic domain size (Figure 4A-C, Figure 4—figure supplement 3, Video 13, Video 14). In later stages, patterns merge and form a large cluster. This type of a large cluster shares visual similarities with our initial swarming experiments (Figure 1A, 1B). Interestingly, unlike simulation the width of the of the cluster reaches saturation before they dissociate. Arrested phase seperation was also observed in bacterial system due to bird and dead dynamics. We noticed that, in our case the consumption of bacteria is significant within the cluster and may limit the growth of the clusters. The similar effect of bacterial depletion was also proposed to explain the motion of the aggregating strain *npr-1*. To investigate the details, we measured the bacterial concentration. GFP signal revealed different bacterial concentrations across the swarm; the front edge of the swarm has more bacteria than the back (Figure 4D-F). Worms in the swarm consumed bacteria and the food continually diffused from the front edge towards the back. As the swarm grows, the gradient profile gradually extends into the swarming body with the average decay length of around λ ~ 2–4 mm (Figure 4G-H). λ the characteristic length scale of the gradient profile can be simply defined by the bacterial diffusion coefficient (D_b_) and consumption rate (f_b_), λ ^2^ = D_b_/f_b_. This decay length defines the width of the swarm body. The small clusters show symmetric GFP distribution and do not form a gradient profile (Figure 4—figure supplement 1D).”